# Implications of Lairage and Environmental Enrichment on Behavioral Responses and Skin Lesions in Finishing Pigs in a Slaughterhouse

**DOI:** 10.3390/ani14111591

**Published:** 2024-05-28

**Authors:** Vivian Schwaab Sobral, Robson Mateus Freitas Silveira, Giovane Debs Guesine, Alessandra Arno, Karen Airosa Machado de Azevedo, Cristian Marcelo Villegas Lobos, Iran José Oliveira da Silva

**Affiliations:** 1Department of Biosystems Engineering, “Luiz de Queiroz” College of Agriculture, University of São Paulo (USP), Piracicaba 13418-900, SP, Brazil; vivianschwaab@gmail.com (V.S.S.); giovane@usp.br (G.D.G.); alessandraarno123@gmail.com (A.A.); karen@usp.br (K.A.M.d.A.); iranoliveira@usp.br (I.J.O.d.S.); 2Department of Animal Science, “Luiz de Queiroz” College of Agriculture, University of São Paulo (USP), Piracicaba 13418-900, SP, Brazil; 3Department of Exact Sciences, University of São Paulo (USP), Piracicaba 13418-900, SP, Brazil; clobos@usp.br

**Keywords:** animal stress, aggressive behavior, metal chains, animal welfare

## Abstract

**Simple Summary:**

Environmental enrichment for pigs in the pre-slaughter room aims to provide a more comfortable and stimulating environment for the animals during the period before slaughter. This can help reduce the stress of transport and improve pig welfare, which can also have positive impacts on reducing the number of animal injuries. The aim of this study was to evaluate the use of environmental enrichment and lairage of pigs in the slaughterhouse on behavioral responses and number of skin lesions. The main results proved that environmental enrichment did not influence skin lesion score or behavioral responses, while the lairage time at slaughter did not influence the number of lesions on the animals. We concluded that environmental enrichment during lairage in a commercial slaughterhouse did not significantly reduce skin lesions or influence the behavior of pigs; that is, the use of simple metal chains as environmental enrichment is not efficient in improving animal welfare in slaughterhouses.

**Abstract:**

Different resources, such as environmental enrichment, are being evaluated in order to minimize animal stress and promote better conditions during the life cycle of animals, as consumers are increasingly concerned about animal welfare issues. Lairage represents an important stage in the swine production chain and is directly related to animal welfare. The aim of this study was to evaluate the effect of lairage time in the slaughterhouse and environmental enrichment on the level of skin lesions and behavioral responses in pigs. A total of 648 finishing pigs of both sexes were assessed before and after lairage at the slaughterhouse with a five-point scale (0 = none, to 4 = ≥16 superficial lesions or >10 deep lesions). After lairage (after slaughter), lesions were also classified according to their source (mounting, fighting, and handling). Pigs were distributed into two treatments groups during lairage: with environmental enrichment (EE) on the pen, with hanging metal chains, and with no enrichment (NE). Behavior was monitored during the first four hours of lairage. Proportional odds, mixed linear model for longitudinal data, and non-parametric Wilcoxon signed rank tests were used to analyze the relation between treatments, skin lesions, and behavior. The simple metal chains did not affect skin lesion score or pigs’ behavior (*p* > 0.05), whereas lairage duration influenced standing (SA), sitting (S), lying (L), idleness (I), and drinking water (D) (*p* < 0.001). The main source of skin lesions was handling, which did not differ between treatments (EE and NE) (*p* > 0.05). It was observed that the severity of the lesions (highest scores of 3, 4, and 5) increased in the different anatomical regions of the pigs when compared before and after slaughter, with the exception of the frontal area, which was the same (*p* = 0.7547). Lairage time has a proportional relation with skin lesions, and hanging chains at the slaughterhouse pens was not enough to reduce the number of lesions and to change pig behavior.

## 1. Introduction

The increasing concern and interest of global markets, associated with regulatory policies, has raised the demand for improvement in processes of animal production to meet animal welfare guidelines [1,2,3] and thus provide humane handling to animals and a protein production chain with high levels of animal welfare [4,5,6]. An important resource to achieve these objectives is the use of environmental enrichment, which can be implemented in different stages of the production cycle [1], as well as during the pre-slaughter operations [7,8,9,10], to reduce the incidence of undesirable behaviors between animals by distracting them and, thus, preventing high levels of stress [11].

Lairage in slaughterhouses is a critical stage in the swine production chain since it can affect animal welfare negatively [12] and cause irreparable damage to the carcass and meat quality, compromising the good practice efforts in previous stages [8] and reducing the opportunity for animals to recover from stress caused by loading, transport, and unloading [13]. Furthermore, it must be considered that poorer meat quality can be associated with fights between pigs, mainly due to transport stress and other factors such as batch mixtures and dominance [14]. Thus, skin lesions can act as an animal welfare [15] and meat quality indicator [16].

Consequently, several approaches to assess animal welfare conditions have been investigated by the scientific community. Slaughter assessments are favorable as they allow for the sampling of a large number of animals per day, reduce the risk of diseases dissemination [17], and reflect the real conditions of the animals’ life cycle, giving more information beyond the pre-slaughter period [15,18].

Pigs have exploratory behavior and when they are not allowed to perform this natural behavior, they tend to react aggressively towards other animals from the pen group [19]. Environmental enrichment in pig farming has been investigated for many years by several authors in order to characterize and define the performance of different materials [20,21]; however, few studies evaluate its use on abattoirs during lairage.

The aim of this study was to evaluate the effect of lairage time in the slaughterhouse and environmental enrichment on the level of skin lesions and behavioral responses in pigs, with the hypothesis that metallic simple chains reduce the incidence of skin lesions as a consequence of the reduction in aggressive behavior between animals, due to greater distraction promoted by the enrichment tools.

## 2. Material and Methods

### 2.1. Animal Housing and Transport

This non-invasive study complies with ARRIVE guidelines (Animal Research: Reporting of In Vivo Experiments) and it was approved by the Ethics Committee on Animal Use (ESALQ/USP protocol number 2019-24).

A total of 648 finishing pigs, containing castrated males and females which were approximately 5 months old and 117 kg of body weight, from 6 different commercial farms located in São Paulo, Brazil, with a distance from slaughterhouses ranging between 135 and 353 km, were assessed. During this experiment, eight groups of pigs were transported on different days, and approximately half of each batch were assessed at the slaughterhouse, totaling 1317 pigs transported during the summer, between February and March 2020.

The pigs were transported in ten pens in two-tiered trailers (five pens per side of the trailer) under a microclimatic variation inside the trailer of 19.8–26.2 °C (temperature), 73.14–92.35% (relative humidity), and 0.35–0.41 kg/m^2^ (stocking density) (kilograms of body weight/m^2^). The vehicles had metallic structures with open sides and grid cross slating floor in aluminum and were conducted by three different drivers in a total of eight journeys.

In all farms, pigs were randomly selected on the day scheduled for slaughter and marked with an individual tattoo number on each side of the hind-quarter in order to enable later identification at the abattoir. Pigs were fasted 12 h before loading and water was provided ad libitum. Farm crews used rattles and plastic tarps to conduct animals to the loading ramp and before departure, pigs were showered, inside the trailer, as a heat dissipation mechanism considering the local temperature.

### 2.2. Unloading and Handling at the Slaughterhouse

Vehicles arrived at the slaughterhouse between 20:50 and 01:00, were weighted before access to the unloading area, and animals were unloaded as soon as possible. Before skin lesion assessment, animals were showered at the arrival pens for cleaning and improving thermal comfort. Injured and fatigued pigs were conducted to an observation pen for further evaluation by the Federal Inspection Service and, thus, disregarded from the experiment, as were dead animals. During the study, 25 pigs were disregarded, of which 23 were moved to observation pens and two were dead on arrival. Figure 1 illustrates unloading, arrival, and lairage areas scheme of the slaughterhouse.

After skin lesion assessment, half of each group of animals were randomly conducted to the conventional lairage pen (NE) and the other half were conducted to the enriched lairage pen (EE), which had non-branched metal chains, equally spaced and hanging in the middle of the pen at a height of 2.0 m with a distance of 30 cm from the floor, distributed in a central row with three units (Figure 2).

Remaining pigs from each batch that were not assessed were also randomly divided into NE and EE pens. All pens had concrete walls and floors with solid stainless-steel gates, 2° slope facing the external corridor to drain waste, and a water sprinkling system in its extension, which was activated for 30 min after the arrival of the animals and 30 min before slaughter. Lairage facilities were supplied with drinking nipples, which provided water ad libitum (Figure 2A,B). Pigs were kept in pens during a regulatory period of 05:00 to 09:00 under similar conditions of stocking density (Table 1).

At lairage, animal behavior was monitored once the NE and EE pens were fully loaded during the first four hours of the resting period by two trained and calibrated observers, who were each responsible for monitoring one of two pens (NE or EE).

Slaughter operations complied with Brazilian regulations, considering the technological procedures and animal welfare requirements. By the end of lairage, pigs were conducted to the stunning area with plastic boards and compressed air jets by the crew of the abattoir and were electrically stunned with a three-point electrode (forehead and heart, 400 V, 1.2A, 6 s, Sulmaq, Rio Grande do Sul, Brazil), inducing cardiac ventricular fibrillation at a restrainer before exsanguination within 30 s.

### 2.3. Data Collection

#### 2.3.1. Skin Lesion Assessment in Pigs

The left side of the pigs’ body was divided into five regions (ear, front, middle, hind- quarter, and limbs) and evaluated according to a five-point scale (0 = none to 4 = ≥16 superficial lesions or >10 deep lesions), adapted from the literature [22,23,24,25], as illustrated by Figure 3. Superficial lesions were defined as skin injuries that did not completely penetrate it (scratches) and deep lesions as gashes or openings that completely penetrate the skin (open wounds) [26]. In order to standardize the evaluation, every 5 cm in length per scratch were counted as five superficial lesions. Due to the fact that they are not related to the objective of this study, skin lesions caused by contact dermatitis, characterized by reddish focal and circular structures, were not included.

Skin lesion score assessment was performed immediately after the unloading of pigs at the slaughterhouse and right before lairage (A1), at the arrival pens. This included the effect of previous operations on the number of skin lesions (handling at farms, loading, transport, and unloading).

#### 2.3.2. Skin Lesion Assessment in Carcasses

Due to the fast speed of the slaughter line, photographic records of the left side of each carcass, identified with the individual tattooed numbers in the hind-quarters, were taken in order to enable the skin lesion assessment. Individual numbers were manually transcribed to the front part in order to improve carcass identification (Figure 4). The evaluation methodology of the skin lesion score of carcasses (A2-after slaughter) was identical, as described in Section 2.3.1. Lesions were also classified by source according to their shape and size as fighting- (≤10 cm length, comma shape and concentrated in the head, front, and hind-quarter parts), mounting- (≥0 cm, comma shape and concentrated on the back, caused by foreclaws) and handling-type (all sizes and irregular shapes, distributed along all parts of the body, caused by rough handling and structures issues), with the latter also classified as “other bruise types”, based on the Institut Technique du Porc [27], represented by Figure 4, and then quantified by their source.

#### 2.3.3. Behavioral Observations during Lairage

The behavior of pigs in EE and NE pens was monitored using the scan sampling method consisting of direct observations at five minutes intervals during the first four hours of lairage, based on ethogram, as represented in Table 2. Animal behaviors were analyzed by three trained technicians with an interobserver reliability of >95%. The proportions of behaviors were calculated according to the total number of pigs per pen and average proportions were obtained per hour. Pigs’ behaviors were divided into two categories; the first covered activities related to the “standing” (SA), “sitting” (SI), and “lying” (L) positions, and the second was related to “idleness” (I), “drinking water” (D), “environment interaction” (EI), “environmental enrichment interaction” (RI), “animal interaction” (AI), and “negative interaction” (NI) activities.

### 2.4. Statistical Analysis

To determine whether or not skin lesion severity for each variable response, represented by body parts (ear, front, middle, hind-quarter and limbs) and evaluated according to the adopted lesion scoring scheme (Figure 3), were influenced by lairage and treatment (EE and NE), a proportional odds model was adopted. Initially, lairage and treatment were tested to evaluate if there was interaction between them. If no effect was observed between them (lairage and treatment), the main effect of each was evaluated by paired comparison tests using the Tukey test. If an interaction was detected, it was divided for each main factor of the model.

The frequency of occurrences of monitored behaviors was assessed using the linear mixed model for longitudinal data in order to test whether or not there was a difference between lairage time, in hours (1 to 4), and treatments (EE and NE). In the model, it was evaluated whether there was interaction between time and treatment, considering the group of animals assessed during hour 4 as a random effect. The variables of study (SA, SI, L, I, D, EI, RI, AI, and NI) were transformed into arcsine of the square root of proportions to achieve model requirements, such as normality and variance homogeneity, among others.

Finally, non-parametric Wilcoxon signed rank test was used to analyze the relation between treatment (EE and NE) and the number of skin lesions caused by fighting, mounting, and handling, based on the carcasses assessment (post-mortem). All statistical analyses were performed in R software, version 4.0.3.

## 3. Results

### 3.1. Skin Lesions: Scoring Scheme and Classification

Considering the five body parts assessed of the pigs and their carcasses (ear, front, middle, hind-quarter, and limbs) at moments A1 (before lairage) and A2 (after lairage, post- mortem), respectively, no interaction was observed between treatment (EE and NE) and lairage; there was also no isolated effect as a result of the use of environmental enrichment with metal chains on skin lesions scores (*p* > 0.05). However, a significant difference was observed between lesions scores in all body parts compared before and after lairage (*p* < 0.001), except for the front part (*p* = 0.7574), indicating that lairage has an important role in the formation of skin lesions (Table 3).

Table 4 shows the numerical proportion of injuries before (A1) and after (A2) slaughter at different points. It was observed that the severity of lesions (highest scores of 3, 4, and 5) increased in different anatomical regions of the pigs when compared before and after slaughter, except for the frontal area, which was the same (*p* = 0.7547)

Through carcass evaluation and skin lesion classification by the source (Figure 5), no significant influence was observed on the number of lesions caused by mounting (*p* = 0.5839), fighting (*p* = 0.8355), and handling (*p* = 0.6692) in relation to lairage treatments (EE and NE) (Table 5), indicating that hanging non-branched metal chains over the extent of lairage pen was not enough to reduce the negative effects on pigs’ skin during lairage at the abattoir significantly. The main source of skin lesions was handling, which presented an average number of 19.76 lesions per carcass from EE lairage and 19.90 from NE lairage, while the mounting-type lesions were the weaker source, which evidenced averages of 0.81 and 0.90 at EE and NE lairage, respectively. Lesions caused by fighting, the second main cause, exhibited average numbers of 1.54 at EE lairage and 1.71 at NE lairage, which, despite having a relevant classificatory position, expressed an evident lower participation in the lesion count in relation to the main source (handling).

The distribution of lesions caused by mounting, fighting, and handling in each part of the carcass (ear, anterior, middle, hind-quarter, and limbs) in relation to the skin injury scores (0, 1, 2, 3, and 4) after lairage under both treatment conditions (EE and NE) were equal (*p* > 0.05) (Figure 5).

### 3.2. Behavior

In this study, there was no interaction between behaviors (SA, SI, L, I, D, EI, RI, AI, and NI) based on treatments (EE and NE) and the monitoring time during lairage, in hours (1 to 4 h). Nevertheless, when evaluating the isolated effect of time and treatment on animals’ behaviors, a significant influence of time was observed (*p* < 0.001), except for the NI (negative interaction) between animals (*p* = 0.0392), which did not significantly vary over time spent in lairage pens. Therefore, treatment did not influence SA, SI, L, I, D, EI, RI, AI, and NI behaviors, indicating that environmental enrichment with hanging non-branched metal chains at the slaughterhouse did not significantly influence pigs’ behavior compared to conventional lairage with no enrichment (Table 6).

## 4. Discussion

In this experiment, the hypothesis that the use of hanging simple metal chains over lairage pens as an environmental enrichment to pigs alters their behavior, including agonistic behaviors, as well as reduces skin lesion levels, was not supported.

In the literature, several studies report a proportional increase in the number of skin lesions with the extension of lairage time [12,16,28,29], which contradicts the objective of this resting period, represented by the stress recovery from previous stages and the regeneration of metabolic sources [16,30,31]. Those findings might be related to a higher incidence of fights between pigs, which may be intensified by the mixing of unfamiliar animals in a new environment [8,29,32], resulting in a greater expression of exploratory behavior and a higher probability of fights occurring between them [32]. Thus, several authors recommend a resting period between 2 and 4 h in order to improve animal welfare conditions [12,16,28,33,34].

Nevertheless, contrary to what was observed in this study, the resting period at the slaughterhouse might not contribute to an increase in skin lesion levels. Panella-Riera et al. [35] did not detect a significant influence of lairage on lesion levels under experimental conditions when considering periods of 0 h and 12 h, justified by the practice of not mixing unfamiliar animals associated with minimal stress pre-slaughter handling conditions. Pérez et al. [36] also did not report a significant effect by lairage durations of 0, 3, and 9 h under commercial conditions on skin lesions.

Peeters and Geers [7], who evaluated the effect of environmental enrichment during lairage on skin lesion levels of pigs, obtained a significant reduction in the incidence of lesions caused by fights in the front part of the animals’ body, which were submitted to two kinds of enrichment—(1) rubber toys and (2) balls filled with maize—in relation to conventional lairage, whereas lesions in the middle and hind-quarter parts did not differ based on treatments. Thus, unlike what was observed in this study, environmental enrichment caused a reduction in skin lesions justified by the alternance of distraction resources, since distinct results were obtained from rubber toys and balls with maize, with the latter being more efficient. Therefore, metal chains might represent a less favorable resource for slaughterhouses, although they are related to better animal welfare conditions in farms when used in bigger sizes and branched, being able to distract two or more pigs simultaneously [37]. On the other hand, no other study that evaluated the use of metal chains during lairage was found to discuss the results of this experiment. Thus, simple metal chains may not be the appropriate environmental enrichment to be provided to pigs during lairage, as they were not able to alter pig behavior, especially pertaining to the reduction in negative interactions. It is known that desirable materials for pig environmental enrichment are edible, chewable, deformable, and searchable ones [38]. Simple metal chains do not meet all these characteristics, indicating a possible reason why our results were not favorable to significantly increasing animal welfare.

Also, as noted in the present study, Peeters and Geers [7] did not find a relation between enrichment treatments and the proportion of animals standing, sitting, and lying, showing significant difference only between animals’ interaction with the types of enrichment tools. Unlike this study, fights were not monitored, but were indirectly referenced by skin lesion score. Investigations made by Faucitano et al. [39] with regard to behavioral differences between pigs with previous contact with environmental enrichment at farms and those that were reared conventionally did not observe a significant effect on the number of lying, drinking water, and fighting animals. Also, Klont et al. [1] did not detect behavioral differences among pigs reared in enriched and conventional rearing systems. In contrast, Barton Gade [40,41] noticed lower aggressive behavior incidences in groups of pigs reared in enriched systems, indicating that animals from the present study could have shown a better conduct if exposed previously to enrichment resources during the productive cycle. In general, those findings corroborate the data presented in this experiment, in which negative interaction frequencies were lower in groups submitted to EE lairage, although not statistically significantly.

Followed by handling, lesions caused by fights were the second main source detected in both lairage treatments (EE and NE) in this study. The mixing of unfamiliar animals during pre-slaughter operations might have provided a higher incidence of negative interaction between them [8,40–42, since pigs were mixed during loading, transport, unloading, and lairage, possibly mitigating the effect of metal chains in the EE pen. In addition, similar to a pattern observed in the literature [40], the chains model (non-branched) may have instigated a competition between pigs over time, considering their novelty to them, since animals were not previously exposed to any enrichment resource, and their availability for individual use.

Lesions caused by reciprocal fights (to establish hierarchy) tend to prevail on the head and front part of the body, while lesions caused by unrequited fights have higher chances of occurring in the hind-quarter of the chased animal [22,30]. Handling-type lesions may occur with higher frequencies in posterior [17,30] and middle parts of the body and, finally, mounting-type lesions can be commonly found in the middle part, especially in the back [30], which may indicate overcrowding [29] and sexual behavior expression [42]. Hence, considering that animals from this experiment presented higher proportions of lesions caused by handling and that the body part with the worst scores (3 and 4) was the middle, it can be inferred that operational procedures and farms, vehicles, and abattoir structures must be revised in order to implement animal welfare improvements, corroborating the possibility of monitoring indirectly such conditions at the slaughter line based on location, shape, and size of skin lesions [15,17,18].

Future perspectives based on our results are the use of a wider range of environmental enrichment strategies, as this study only used simple metal chains. It is known that exploring different types of environmental enrichment and their effects on pig behavior and welfare can offer a more comprehensive understanding of how environmental factors affect pigs during lairage. Furthermore, we recommend carrying out longitudinal studies to assess the long-term effects of enrichment on pig welfare, as this could provide valuable information to improve animal welfare practices in slaughterhouses.

## 5. Conclusions

Pre-slaughter operations influenced the number of lesions on the skin of pigs, with handling being the main cause of serious lesions in animals. Furthermore, the time spent by animals in housing influences behavioral responses and the number of injuries specifically, increasing the number of injuries, with scores 3, 4, and 5, in different parts of the body, except for the animal’s forehead. It is also concluded that hanging unbranched metal chains along the length of the corral as environmental enrichment does not reduce the number of skin injuries or the animals’ behavior.

## Figures and Tables

**Figure 1 animals-14-01591-f001:**
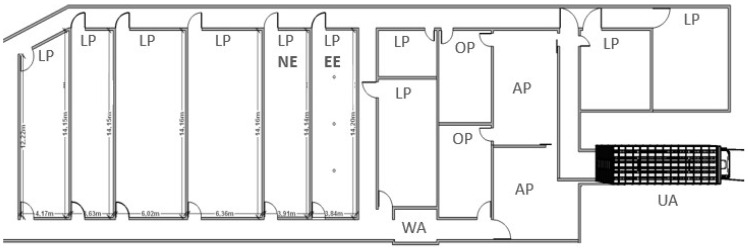
Schematic presentation of unloading area (UA), arrival pens (AP), observation pens (OP), weighing area (WA), and lairage pens (LP) of the slaughterhouse. Marked letters represent conventional lairage (no environmental enrichment), pen (NE), and enriched lairage pen (EE) selected for the study, the latter with the graphical representation by the small circles of the three hanging chains on the extension of the pen.

**Figure 2 animals-14-01591-f002:**
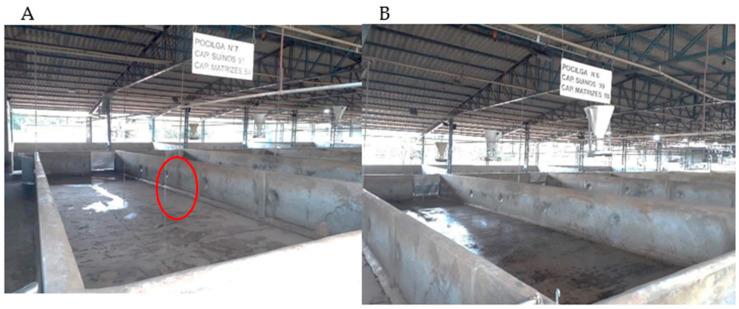
(**A**) Lairage pen with environmental enrichment (EE), with a row of three metal chains and (**B**) conventional lairage pen, with no environmental enrichment (NE). Note: Red circle shows non-branched metal chains.

**Figure 3 animals-14-01591-f003:**
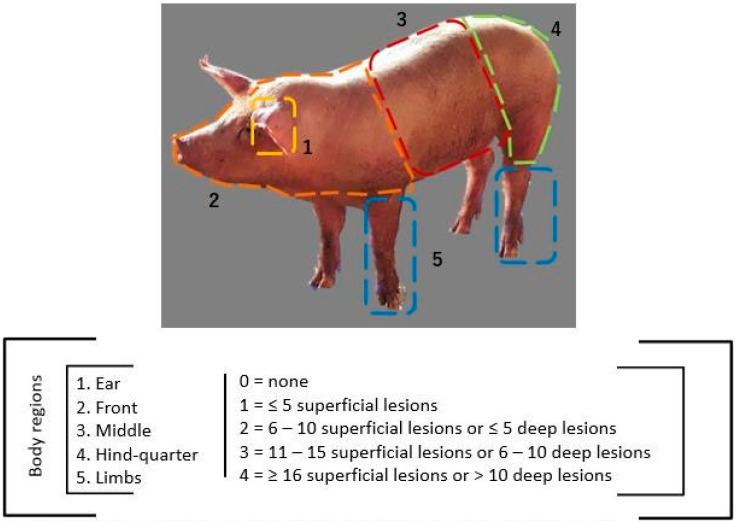
Lesion scoring scheme of pigs at the left side of the body, divided into 5 regions: (1) ear; (2) front, considering the head to back of shoulder; (3) middle, back of shoulder to hind-quarter; (4) hind-quarter; and (5) limbs. Adapted from the literature [22,23,24,25].

**Figure 4 animals-14-01591-f004:**
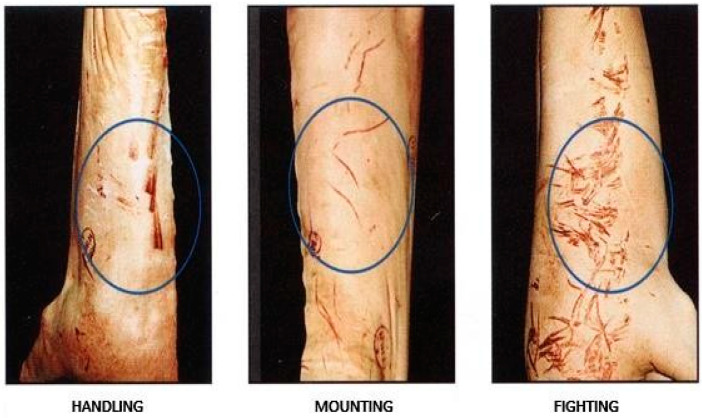
Photographic standards of the Institut Technique du Porc [27]. Note: Circle represents region of injuries depending on body region.

**Figure 5 animals-14-01591-f005:**
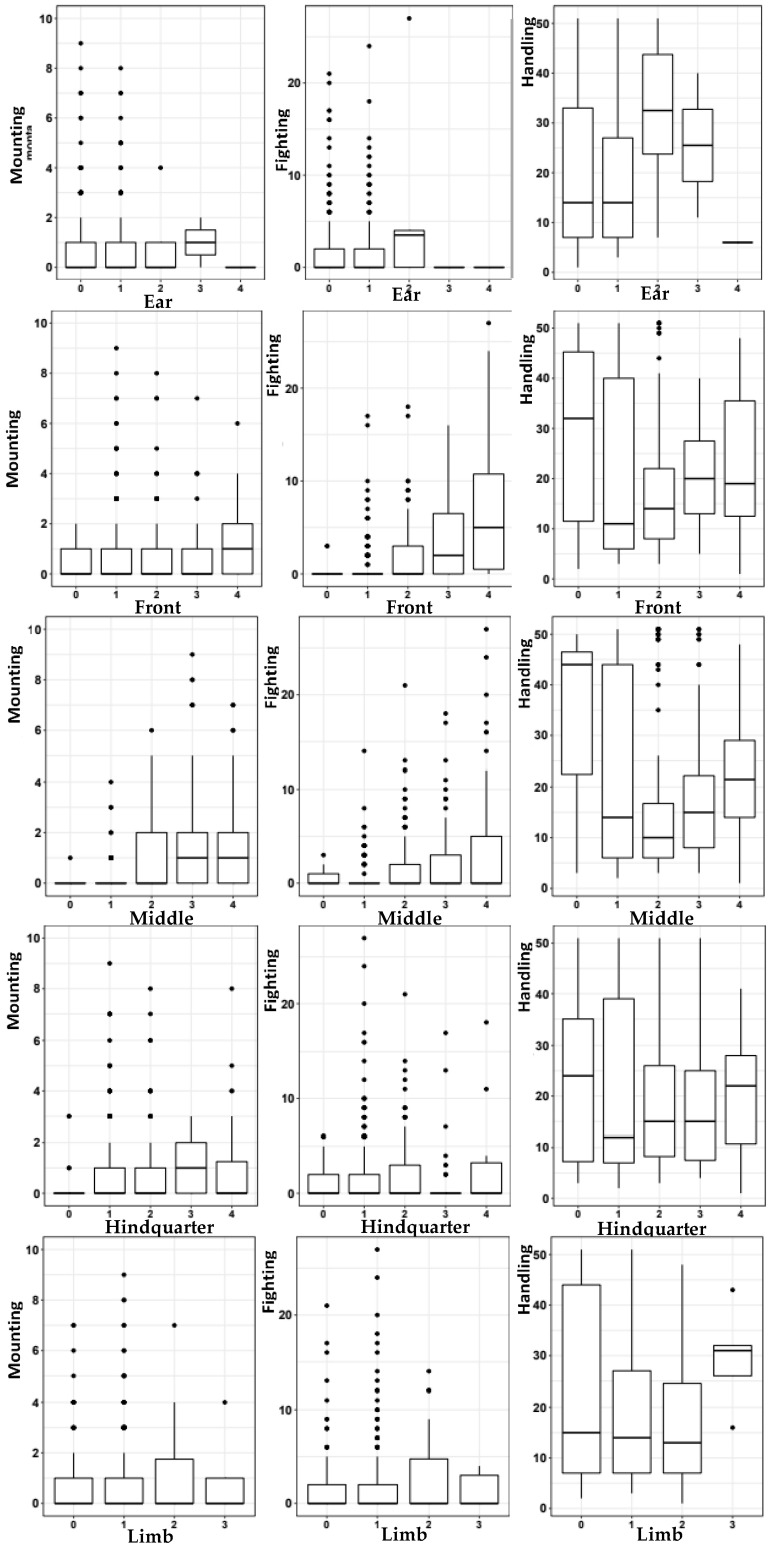
Distribution of classified lesions according to the source based on the scores of skin lesions on carcasses.

**Table 1 animals-14-01591-t001:** Total number of pigs per transport group, assessed and disregarded (deviated and dead); lairage duration and stocking density (m^2^/110 kg of living weight) from each batch of animals of this experiment.

Day of Transport (Batch)	Number of Assessed Pigs	Number of Pigs		Number of Assessed Pigs	Lairage (h)	Stocking Density (m^2^/110 kg) at Lairage	
		Dead at Arrival	Deviated			Enriched (EE)	Non Enriched (NE)
1	160	0	0	80	09:00	0.64	0.63
2	160	0	1	80	08:15	0.68	0.68
3	165	0	0	80	08:50	0.62	0.62
4	160	0	6	80	07:55	0.69	0.68
5	160	0	8	80	05:10	0.72	0.71
6	160	1	6	80	07:00	0.70	0.70
7	176	1	0	84	05:00	0.64	0.63
8	176	0	0	84	06:25	0.64	0.63
**Total**	1317	2	21	648			

**Table 2 animals-14-01591-t002:** Behavior ethogram of pigs during lairage with (EE) and without environmental enrichment (NE).

Behavior	Description
Standing (SA)	Pig on all four extended members, stationary or walking
Sitting (SI)	Pig with rump on the ground with front legs straight
Lying (L)	Pig with all ventral or lateral body surface- supported on the floor
Idleness (I)	Pig performing no activity
Drinking water (D)	Pig drinking water from drinking nipples
Environment interaction (EI)	Pig exploring pen floor, walls and gates
Environmental enrichment interaction (RI)	Pig sniffing, biting or sucking the metal chains
Animal interaction (AI)	Pig exploring ears, tails or belly from another- pig with the nose
Negative interaction (NI)	Pig biting (flank, neck, tail, head and ears),— sucking the belly, mounting and/or scratching another pig

**Table 3 animals-14-01591-t003:** Average values of interactions obtained between lairage and treatment (EE and NE) and the isolated effects of lairage and treatment on pig and carcass skin lesion score on the five body parts evaluated (ear, front, middle, hind-quarter, and limbs).

Body Parts	Treatment	Lairage	Interaction (Treatment × Lairage)
Ear	0.2399	<0.001	0.3833
Front	0.9165	0.7547	0.1387
Middle	0.2712	<0.001	0.0716
Hind-quarter	0.9321	<0.001	0.2779
Limbs	0.3979	<0.001	0.6526

**Table 4 animals-14-01591-t004:** Proportion of pigs with lesions scores (0 to 4) on the five body parts (ear, front, middle, hind-quarter, and limbs) before (A1) and after lairage (A2) with (EE).

Regions		Scores				
		0	1	2	3	4
Ear	A1	0.419	0.499	0.072	0.008	0.001
	A2	0.595	0.393	0.010	0.002	0.001
Middle	A1	0.061	0.515	0.286	0.092	0.045
	A2	0.031	0.554	0.284	0.074	0.057
Front	A1	0.067	0.513	0.280	0.094	0.045
	A2	0.012	0.290	0.331	0.213	0.153
Hind-quarter	A1	0.168	0.560	0.186	0.059	0.027
	A2	0.049	0.575	0.260	0.078	0.039
Limbs	A1	0.419	0.485	0.082	0.004	0.01
	A2	0.249	0.673	0.067	0.010	0.01

**Table 5 animals-14-01591-t005:** Average number of lesions caused by mounting, fighting, and handling on swine carcasses after conventional and enriched lairage, evaluated by Wilcoxon signed rank test.

Lesion’s Source	Average Number of Skin Lesions on Carcasses		Treatments (EE e NE)	
	Enriched Lairage (EE)	Conventional Lairage (NE)	Wilcoxon Test	*p*-Value
Mounting	0.81	0.90	40728	0.5839
Fighting	1.54	1.71	41361	0.8355
Handling	19.76	19.90	42558	0.6692

**Table 6 animals-14-01591-t006:** Proportions per hour of animals’ behaviors from enriched (EE) and conventional (NE) lairage during observation time by scan sampling (4 h) followed by the significance of interaction between time and lairage treatment (EE and NE) for each behavior and its isolated effects over behaviors (SA, SI, L, I, D, EI, RI, AI, and NI) ^1^.

		Behavior ^1^ (%)								
Lairage	Time (h)	SA	SI	L	I	D	EI	RI	AI	NI
EE	1	16.83	2.10	81.06	84.89	0.69	5.18	3.85	4.97	0.61
	2	1.38	1.01	97.61	97.16	0.05	0.53	0.50	1.09	0.57
	3	1.98	1.17	96.85	96.83	0.04	0.78	0.71	1.10	0.46
	4	3.62	1.49	94.89	95.74	0.13	1.01	1.18	1.47	0.31
NE	1	20.08	3.19	76.73	85.88	0.83	5.05	NA	6.75	1.31
	2	3.25	0.80	95.95	96.91	0.10	0.85	NA	1.05	1.09
	3	1.50	0.86	97.64	98.71	0.05	0.39	NA	0.65	0.20
	4	1.24	0.79	97.96	98.95	0.04	0.21	NA	0.59	0.20
	***p*-value**									
		**SA**	**SI**	**L**	**I**	**D**	**EI**	**RI**	**AI**	**NI**
(time × treatment)		0.3521	0.1584	0.5206	0.8939	0.6033	0.4247	NA	0.4221	0.5793
Time		<0.001	<0.001	<0.001	<0.001	<0.001	<0.001	NA	<0.001	0.0392
Treatment		0.0559	0.9985	0.7344	0.5455	0.9727	0.2650	NA	0.7015	0.6805

^1^: “Standing” (SA), “Sitting” (SI), “Lying” (L), “Idleness” (I), “Drinking water” (D), “Environment interaction” (EI), “Environmental enrichment interaction” (RI), “Animal interaction” (AI), and “Negative interaction” (NI); NA: Not applicable.

## Data Availability

The datasets generated and/or analyzed during the current study are available upon request to the corresponding author.

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
