# Peer review of "Implications of Lairage and Environmental Enrichment on Behavioral Responses and Skin Lesions in Finishing Pigs in a Slaughterhouse"

_animals, 2024, doi:10.3390/ani14111591_

Round 1
Reviewer 1 Report
Comments and Suggestions for Authors
The use of environmental enrichment can reduce the incidence of undesirable behaviors and prevent high levels of stress for animals. The manuscript of “The use of environmental enrichment during lairage and its reflects on behaviour and skin lesions in finishing pigs” from Dr. Sobral et., al found that environmental enrichment during lairage at the slaughterhouse neither significantly reduced skin lesions nor influenced pig’s behavior responses, but the time spent by animals during lairage can significantly influence the behavior and the amount of skin lesions in different parts of their body. The manuscript was well-written and the authors did well in many of their experiments and their interpretations. It can be accepted after minor revision taking into consideration the following points.
1. Figure 5. was Photographic standards of the Institut Technique du Porc (ITP, 1996), can you indicate and quantify of lesions by the source for fighting-, mounting- and handling like figure 4. Because the standards lesions was not clear and obvious than picture 4.
2. All the behaviors are recorded using the percentage of observation time but not using the frequency. Can behavioral observations be described in more detail?
3. Figure 7. Distribution of classified lesions according to the source based on the scores of skin lesions scores on carcasses. The results were not significance. Whether statistical analysis by mixing these influencing factors(time spend, treatments, lairage, carcasses parts et., al) will yield significantly different results?
Author Response
#Reviewer 1
The use of environmental enrichment can reduce the incidence of undesirable behaviors and prevent high levels of stress for animals. The manuscript of “The use of environmental enrichment during lairage and its reflects on behaviour and skin lesions in finishing pigs” from Dr. Sobral et., al found that environmental enrichment during lairage at the slaughterhouse neither significantly reduced skin lesions nor influenced pig’s behavior responses, but the time spent by animals during lairage can significantly influence the behavior and the amount of skin lesions in different parts of their body. The manuscript was well-written and the authors did well in many of their experiments and their interpretations. It can be accepted after minor revision taking into consideration the following points.
- Figure 5. was Photographic standards of the Institut Technique du Porc (ITP, 1996), can you indicate and quantify of lesions by the source for fighting-, mounting- and handling like figure 4. Because the standards lesions was not clear and obvious than picture 4.
Response: The figures had their titles changed! Figure 5 was used as a standard for classifying animals according to injuries. Therefore, the other figure was excluded.
- All the behaviors are recorded using the percentage of observation time but not using the frequency. Can behavioral observations be described in more detail?
Response: Review
The behavior of pigs in EE and NE pens was monitored using the scan sampling method consisting of direct observations at each 5 minutes intervals along the first 4 hours of lairage, based on ethogram represented by Table 2. Three trained technicians analyzed the behaviors with interobserver reliability > 95%. The proportions of behaviors were calculated according to the total number of pigs per pen and average proportions were obtained by hour
Reviewer 2 Report
Comments and Suggestions for Authors
The article is a comprehensive document that presents interesting data related to the " The effect of environmental enrichment during lairage and lairage time on behaviour and skin lesions in finishing pigs”. Although the topic falls within the scope of Animals, from my perspective the study has major limitations that make it impossible to publish it at the moment. Of these, I would emphasise:
- Simple summary: The authors mention transport time and its influence on skin lesions. This was not the subject of the study, so the sentences should be reviewed
- Lairage time was not included in the title and objectives, but was also studied and had a more relevant effect on lesions score and behavior comparatively to enrichment material. For that reason, it should be included in the title and objectives
- The enrichment material used was not appropriate for animal welfare. It is now known that enrichment material must be edible, chewable, investigable and manipulable. Metal chains do not fulfil all these characteristics. For this reason, the authors conclude in this study that it seems that metal chains may represent a less favourable resource for slaughterhouses. Based on other studies, the authors report that rubber toys and corn balls seem to be more efficient.
- In line 236-237, authors describe that “.. a significant difference was observed between lesions scores in all body parts comparing before and after lairage (p<0.001), …, indicating that lairage has an important role in skin lesions formation”. But, in line 255-256, the authors refer that there was no significant difference between the proportions of skin lesion scores based on assessment times A1 (after unloading, before transport) and A2 (post-mortem). Furthermore, figure 6 is not clear to determine the differences between lesions scores before and after lairage. If they are not different, it means that the lesions observed were the result of stages prior to the lairage (transport....), making it impossible to use them for studies related to their association with time and enrichment material in the lairage. If this is not the case, the authors will have to demonstrate how they accounted for the injuries that resulted only from their stay in the lairage. These results should be the first to be clearly presented in a table (not in a graphic) in the results chapter. Until this matter is clarified, I will not be able to accept the article for publication

Comments on the Quality of English LanguageMinor editing of English language required
Author Response
#Reviewer 2
The article is a comprehensive document that presents interesting data related to the " The effect of environmental enrichment during lairage and lairage time on behaviour and skin lesions in finishing pigs”. Although the topic falls within the scope of Animals, from my perspective the study has major limitations that make it impossible to publish it at the moment. Of these, I would emphasise:
- Simple summary: The authors mention transport time and its influence on skin lesions. This was not the subject of the study, so the sentences should be reviewed
Response: Reviewed
Simple Summary: Environmental enrichment in pigs in the pre-slaughter room aims to provide a more comfortable and stimulating environment for the animals during the period before slaughter. This can help reduce the stress of transport and improve pig welfare, which can also have positive impacts on reducing the number of animal injuries. The objective of this study was to evaluate the use of environmental enrichment and the lairage time of pigs in the slaughterhouse on behavioral responses and the number of skin lesions. The main results were that enrichment did not influence the skin lesion score, the lairage time in the slaughterhouse influenced the number of lesions on the animals' skin. We concluded that environmental enrichment during housing in a commercial slaughterhouse did not significantly reduce skin lesions or influence the behavior of pigs, that is, the use of chains as environmental enrichment is not efficient in improving animal welfare in slaugh-terhouses.
- Lairage time was not included in the title and objectives, but was also studied and had a more relevant effect on lesions score and behavior comparatively to enrichment material. For that reason, it should be included in the title and objectives
Response:
Title: Implications of lairage time in the slaughterhouse and envi-ronmental enrichment on behavioral responses and skin lesions in finishing pigs
Objectives: The objective of this study was to evaluate the effect of lairage time in the slaughterhouse and environmental enrichment on the level of skin lesions and behavioral responses in pigs
- The enrichment material used was not appropriate for animal welfare. It is now known that enrichment material must be edible, chewable, investigable and manipulable. Metal chains do not fulfil all these characteristics. For this reason, the authors conclude in this study that it seems that metal chains may represent a less favourable resource for slaughterhouses. Based on other studies, the authors report that rubber toys and corn balls seem to be more efficient.
Response: Reviewed
Thanks for the comment! I put it as a limitation of the study!
On the other hand, no other study which evaluated the use of metal chains during lairage was found to discuss the results of this experiment. Therefore, one of the limitations of this study may have been that iron chains were used as environmental enrichment, as they were not the most suitable enrichment material for animal welfare. Recently it has been argued that the best materials for environmental enrichment are materials that are edible, chewable, researchable and manipulable (Smith and Pierdon, 2024). Metal chains do not meet all of these characteristics; Possibly, our results were not favorable for increasing animal welfare. We encourage future studies with environmental enrichment in the slaughterhouse using rubber toys and corn balls, which appear to be more efficient.
- In line 236-237, authors describe that “.. a significant difference was observed between lesions scores in all body parts comparing before and after lairage (p<0.001), …, indicating that lairage has an important role in skin lesions formation”. But, in line 255-256, the authors refer that there was no significant difference between the proportions of skin lesion scores based on assessment times A1 (after unloading, before transport) and A2 (post-mortem).
Response: Reviewed
The sentence was rewritten.
Furthermore, figure 6 is not clear to determine the differences between lesions scores before and after lairage. If they are not different, it means that the lesions observed were the result of stages prior to the lairage (transport....), making it impossible to use them for studies related to their association with time and enrichment material in the lairage. If this is not the case, the authors will have to demonstrate how they accounted for the injuries that resulted only from their stay in the lairage. These results should be the first to be clearly presented in a table (not in a graphic) in the results chapter. Until this matter is clarified, I will not be able to accept the article for publication
Figure 6 was deleted and replaced by a table and its results were presented and discussed!
Reviewer 3 Report
Comments and Suggestions for Authors
This study looked at the influence of environmental enrichment on pig lesions and behavior before slaughter. The authors compared pigs in enriched lairages and typical ones to determine if EE would improve pig welfare, represented by fewer lesions. The study is valuable given the need to improve animal welfare, but parts of the paper, as written, include much detail, and some sections need more. The introduction lacks information on why this study is needed despite “several approaches to monitor animal welfare” that currently exist. They can also add more information about the specific goals of the chain EE and how it is intended to improve welfare.
Below are some specific notes.
Line 41: Global markets
Line 43: processes involved in what? Add a word or two to tie it together.
Line 46: missing brackets
Line 51: Define lairage clearly.
Line 56-57: the statement “damages caused in pigs” is unclear, can you be more specific, or even so, this may not be needed here.
Line 59: should this be “established” if so this sentence also needs more detail, what do you mean by types of behaviors established?
Line 62: by the scientific community
Line 64: an animal life cycle
Line 65: beyond the pre-slaughter period
Line 74: What does ARRIVE stand for?
Line 76: who was approximately 5 months old.
Line 77: add a space between numbers and units (e.g. 117 kg)
Line 84: add superscript for meters.
Line 102: clinical alterations such as…
Table one: the text notes two pigs were dead on arrival, but the table only shows 1
Figure 2: the EE in the image is unclear. What is the goal of the chain EE?
Line 222: you can write out numbers less than 10, such as five in the instance.
Figure 7: the authors should explain why they chose to include the outliers.
Were the number of lesions recorded before they entered the two treatments, is it possible that they got these lesions in transport, which would skew the way this data can be interpreted if they are not all starting with no or a comparable number of lesions.
Line 344: In the literature,
Line 406-423: check for consistency the citations are now in all caps.
Line 433: change could not be proven to “was not supported.”
Comments on the Quality of English Languagesufficent
Author Response
#Reviewer 3
This study looked at the influence of environmental enrichment on pig lesions and behavior before slaughter. The authors compared pigs in enriched lairages and typical ones to determine if EE would improve pig welfare, represented by fewer lesions. The study is valuable given the need to improve animal welfare, but parts of the paper, as written, include much detail, and some sections need more. The introduction lacks information on why this study is needed despite “several approaches to monitor animal welfare” that currently exist. They can also add more information about the specific goals of the chain EE and how it is intended to improve welfare.
Below are some specific notes.
Line 41: Global markets
Line 43: processes involved in what? Add a word or two to tie it together.
Response: Reviewed
The increasing concern and interest of global markets, associated with the regulatory policies, has raised the demand for improvement of the processes in animal production along with animal welfare guidelines (Klont et al., 2001; Brandt et al., 2017, Nannoni et al., 2023),
Line 46: missing brackets
Response: Reviewed
An important resource to achieve these objectives is the use of environmental enrichment (Van de Weerd & Ison, 2019)
Line 51: Define lairage clearly.
Response: Reviewed
Lairage time in the slaughterhouse is a critical stage in the swine production chain since it can negatively affect animal welfare (Čobanović et al., 2016a)
Line 56-57: the statement “damages caused in pigs” is unclear, can you be more specific, or even so, this may not be needed here.
Response: Reviewed
Furthermore, it must be considered that poorer meat quality can be associated with fights between pigs, mainly due to transport stress and other factors such as batch mixtures and dominance (Sobral et al., 2024).
Line 59: should this be “established” if so this sentence also needs more detail, what do you mean by types of behaviors established?
Response: Sentence removed
Line 62: by the scientific community
Response: Reviewed
Line 64: an animal life cycle
Response: Reviewed
Line 65: beyond the pre-slaughter period
Response: Reviewed
Line 74: What does ARRIVE stand for?
Response: Reviewed
This non-invasive study complies with ARRIVE guidelines (Animal Research: Reporting of In Vivo Experiments) and it was approved by the Ethic Committee on Animal Use (ESALQ/USP protocol number 2019-24).
Line 76: who was approximately 5 months old.
Line 77: add a space between numbers and units (e.g. 117 kg)
Response 76-77 : Reviewed
who was approximately 5 months old and 117 kg of body weight
Line 84: add superscript for meters.
Response: Reviewed
Line 102: clinical alterations such as…
Response: Reviewed
Table 1: the text notes two pigs were dead on arrival, but the table only shows 1
Response: Reviewed
Figure 2: the EE in the image is unclear. What is the goal of the chain EE?
Response: Reviewed
Add:
Note: Red circle representing iron chains
Line 222: you can write out numbers less than 10, such as five in the instance.
Response: Reviewed
Figure 7: the authors should explain why they chose to include the outliers.
Response: These values are not outliers. We actually observed these occurrences in the field. For this reason we leave the raw values! It is common in data of this nature for occurrences of this dispersion!
Were the number of lesions recorded before they entered the two treatments, is it possible that they got these lesions in transport, which would skew the way this data can be interpreted if they are not all starting with no or a comparable number of lesions.
Thanks for the sugestion! It's an excellent observation! However, we don't have this data! We can consider it in future studies!
Line 344: In the literature,
Response: Reviewed
Line 406-423: check for consistency the citations are now in all caps.
Response: Reviewed
Line 433: change could not be proven to “was not supported.”
Response: Reviewed
Reviewer 4 Report
Comments and Suggestions for Authors
The paper titled "The use of environmental enrichment during lairage and its effects on behavior and skin lesions in finishing pigs" aims to evaluate the impact of environmental enrichment on skin lesions and behavior of pigs during lairage. The study found that hanging metal chains as enrichment did not significantly affect skin lesions or behavior, but lairage duration influenced various behaviors. The main source of skin lesions was handling, not affected by enrichment. Lairage time correlated with skin lesions, suggesting that hanging chains were insufficient to reduce lesions or alter pig behavior.
The research addresses the important topic of animal welfare in swine production, specifically focusing on the impact of environmental enrichment during lairage on pigs' well-being. It contributes valuable insights into the relationship between environmental factors and animal welfare, highlighting the need for further research in this area. The study adds to existing literature by specifically examining the effects of hanging metal chains as enrichment during lairage, providing new data on this aspect of pig management.
I recommend rewriting the abstract and including more results and the significance of the obtained data.
To enhance the research's appeal, I suggest avoiding the inclusion of terms in the keywords that are already present in the article title.
The topic of improving the welfare conditions of pigs is becoming increasingly central, especially given its importance to consumers. To this end, researchers have evaluated various strategies, including the use of different types of environmental enrichments. For further insights, I recommend reading and citing the paper: 10.3390/ani13182967.
Lines 41-43: I suggest citing 10.1016/j.rvsc.2023.03.008 to provide an example from another farm animal category.
Lines 54-55: The meat modifications occurring during the stressful transportation to the slaughterhouse are reported in 10.3390/ani10122386 and 10.3390/ani10060945. Please refer to these articles to add more information to your manuscript.
In terms of methodology, the authors should consider further controls to enhance the study's robustness. Specifically, improvements in monitoring and controlling variables related to behavior and skin lesions could strengthen the research findings. Additionally, a more detailed analysis of the interactions between environmental enrichment and pig behavior could provide deeper insights into the effectiveness of enrichment strategies.
Future perspectives for the authors could involve expanding the study to include a broader range of environmental enrichment strategies beyond hanging metal chains. Exploring different types of enrichment and their effects on pig behavior and welfare could offer a more comprehensive understanding of how environmental factors impact pigs during lairage. Additionally, considering longitudinal studies to assess the long-term effects of enrichment on pig well-being could provide valuable insights for improving animal welfare practices in swine production.
The conclusions drawn in the paper should be carefully evaluated to ensure they align with the evidence presented. It is essential to verify that the conclusions adequately address the main research question posed and are supported by the arguments and data provided in the study. Furthermore, a thorough review of references is crucial to ensure their relevance and accuracy in supporting the research findings.
The authors of the manuscript should consider future perspectives for disseminating their results, particularly focusing on using social media to reach a wider audience. Recent publications emphasize the importance of leveraging social media platforms to share research findings effectively and engage with a broader community. By utilizing platforms like Twitter, ResearchGate, or LinkedIn, the authors can increase the visibility and impact of their study, facilitating knowledge dissemination and fostering discussions within the scientific community. See: 10.3390/ani13223503
Moreover, it is essential for the authors to ensure that their reference list is comprehensive and accurately reflects all citations made in the main text. Double-checking the alignment between references cited in the manuscript and those listed in the reference section is crucial to maintain academic integrity and provide readers with access to the sources supporting the research.
In terms of specific comments on the manuscript, it would be beneficial for the authors to address any potential shortcomings or errors related to data analysis, interpretation of results, or clarity of presentation. Providing constructive feedback on areas that require revision or clarification can enhance the overall quality and impact of the study. Additionally, ensuring that all key points are clearly articulated and supported by evidence is essential for strengthening the manuscript's credibility and relevance in the field.
Lastly, authors should consider discussing any additional points that may need attention in their manuscript. This could include addressing any limitations of the study, proposing further research directions based on the current findings, or highlighting implications for practical applications in swine production. By thoroughly examining these aspects, the authors can enhance the depth and significance of their research contribution.
Author Response
The paper titled "The use of environmental enrichment during lairage and its effects on behavior and skin lesions in finishing pigs" aims to evaluate the impact of environmental enrichment on skin lesions and behavior of pigs during lairage. The study found that hanging metal chains as enrichment did not significantly affect skin lesions or behavior, but lairage duration influenced various behaviors. The main source of skin lesions was handling, not affected by enrichment. Lairage time correlated with skin lesions, suggesting that hanging chains were insufficient to reduce lesions or alter pig behavior.
The research addresses the important topic of animal welfare in swine production, specifically focusing on the impact of environmental enrichment during lairage on pigs' well-being. It contributes valuable insights into the relationship between environmental factors and animal welfare, highlighting the need for further research in this area. The study adds to existing literature by specifically examining the effects of hanging metal chains as enrichment during lairage, providing new data on this aspect of pig management.
I recommend rewriting the abstract and including more results and the significance of the obtained data.
Response: Reviewed
Simple Summary: Environmental enrichment in pigs in the pre-slaughter room aims to provide a more comfortable and stimulating environment for the animals during the period before slaughter. This can help reduce the stress of transport and improve pig welfare, which can also have positive impacts on reducing the number of animal injuries. The objective of this study was to evaluate the use of environmental enrichment and the lairage time of pigs in the slaughterhouse on behavioral responses and the number of skin lesions. The main results were that enrichment did not influence the skin lesion score, the lairage time in the slaughterhouse influenced the number of lesions on the animals' skin. We concluded that environmental enrichment during housing in a commercial slaughterhouse did not significantly reduce skin lesions or influence the behavior of pigs, that is, the use of chains as environmental enrichment is not efficient in improving animal welfare in slaughterhouses.
Abstract: Different resources are being evaluated in order to minimize animal stress and to proportion better conditions during life cycle, such as environmental enrichment, once consumers are more concerned about animal welfare issues. Lairage represents an important stage of swine production chain and it’s directly related to animal welfare. The objective of this study was to evaluate the effect of lairage time in the slaughterhouse and environmental enrichment on the level of skin lesions and behavioral responses in pigs. 648 finishing pigs of both sexes were assessed before and after lairage at the slaughterhouse with a five-point scale (0= none to 4= ≥16 superficial lesions or >10 deep lesions). After lairage (after slaughter), lesions were also classified according to its source (mounting, fighting and handling). Pigs were distributed into 2 treatments groups during lairage: with environmental enrichment (EE) on the pen, with hanging metal chains, and without enrichment (NE). Behavior was monitored during the first 4 hours of lairage. Proportional odds; mixed linear model for longitudinal data and non-parametric Wilcoxon signed rank test were used to analyze the relation between treatments and skin lesions and behavior. The chains didn’t affect skin lesion score neither pig’s behavior (p>0.05), whereas lairage duration influenced standing (SA), sitting (S), lying (L), idleness (I) and drinking water (D)(p<0.001). The main source of skin lesions was handling, which didn’t differ between treatments (EE and NE) (p>0.05). Higher proportions of scores 2, 3 and 4 were observed at posterior part of the body at treatment EE. Lairage time has a proportional relation with skin lesion and hanging chains at the slaughterhouse pens wasn’t enough to reduce the number of lesions and to change pig behavior.
To enhance the research's appeal, I suggest avoiding the inclusion of terms in the keywords that are already present in the article title.
Response: Reviewed
Keywords: animal stress, aggressive behavior, metal chains, welfare animal
The topic of improving the welfare conditions of pigs is becoming increasingly central, especially given its importance to consumers. To this end, researchers have evaluated various strategies, including the use of different types of environmental enrichments. For further insights, I recommend reading and citing the paper: 10.3390/ani13182967.
Response>
Lines 41-43: I suggest citing 10.1016/j.rvsc.2023.03.008 to provide an example from another farm animal category.
The increasing concern and interest of global markets, associated with the regulatory policies, has raised the demand for improvement of the processes in animal production along with animal welfare guidelines (Klont et al., 2001; Brandt et al., 2017, Nannoni et al., 2023)
Lines 54-55: The meat modifications occurring during the stressful transportation to the slaughterhouse are reported in 10.3390/ani10122386 and 10.3390/ani10060945. Please refer to these articles to add more information to your manuscript.
Response: Reviewed
An important resource to achieve these objectives is the use of environmental enrichment (Van de Weerd & Ison, 2019) which can be implemented in different stages of the production cycle (Klont et al., 2001), as well as during the pre-slaughter operations (Peeters; Geers, 2006; Faucitano et al., 2020, Sardi et al., 2020ab),
Future perspectives for the authors could involve expanding the study to include a broader range of environmental enrichment strategies beyond hanging metal chains. Exploring different types of enrichment and their effects on pig behavior and welfare could offer a more comprehensive understanding of how environmental factors impact pigs during lairage. Additionally, considering longitudinal studies to assess the long-term effects of enrichment on pig well-being could provide valuable insights for improving animal welfare practices in swine production.
Response: Reviewed
Line 440- 446: Future perspectives based on our results are the use of a wider range of environmental enrichment strategies, as this study only used metallic chains. It is known that exploring different types of enrichment and their effects on pig behavior and welfare can offer a more comprehensive understanding of how environmental factors affect pigs during housing. Furthermore, we recommend carrying out longitudinal studies to assess the long-term effects of enrichment on pig welfare, as this could provide valuable information to improve animal welfare practices in pig production in slaughterhouses.
The conclusions drawn in the paper should be carefully evaluated to ensure they align with the evidence presented. It is essential to verify that the conclusions adequately address the main research question posed and are supported by the arguments and data provided in the study. Furthermore, a thorough review of references is crucial to ensure their relevance and accuracy in supporting the research findings.
Response: Reviewed
Moreover, it is essential for the authors to ensure that their reference list is comprehensive and accurately reflects all citations made in the main text. Double-checking the alignment between references cited in the manuscript and those listed in the reference section is crucial to maintain academic integrity and provide readers with access to the sources supporting the research.
Response: Reviewed
In terms of specific comments on the manuscript, it would be beneficial for the authors to address any potential shortcomings or errors related to data analysis, interpretation of results, or clarity of presentation. Providing constructive feedback on areas that require revision or clarification can enhance the overall quality and impact of the study. Additionally, ensuring that all key points are clearly articulated and supported by evidence is essential for strengthening the manuscript's credibility and relevance in the field. Lastly, authors should consider discussing any additional points that may need attention in their manuscript. This could include addressing any limitations of the study, proposing further research directions based on the current findings, or highlighting implications for practical applications in swine production. By thoroughly examining these aspects, the authors can enhance the depth and significance of their research contribution.
Response: Reviewed
On the other hand, no other study which evaluated the use of metal chains during lairage was found to discuss the results of this experiment. Therefore, one of the limitations of this study may have been that iron chains were used as environmental enrichment, as they were not the most suitable enrichment material for animal welfare. Recently it has been argued that the best materials for environmental enrichment are materials that are edible, chewable, researchable and manipulable (Smith and Pierdon, 2024). Metal chains do not meet all of these characteristics; Possibly, our results were not favorable for increasing animal welfare. We encourage future studies with environmental enrichment in the slaughterhouse using rubber toys and corn balls, which appear to be more efficient.
Reviewer 5 Report
Comments and Suggestions for Authors
General comment: This study evaluated the use of chains in the lairage pen on the lesions and and behaviours of pigs. Despite not finding a significant effect of the chains on any of the variables investigated, I believe that this study would deserve to be published, as the work was well-designed and conducted.
However, significant improvements must be made in the writing of the manuscript before its publications:
- avoid using contractions in the negative form (don't, wasn't). This is grammatically incorrect to use them in formal writing, such as in a research paper
- Number under 10 (that are not measures) should be written in full
- some sections would benefit from clearer wording
- the results section should not contain allegation on differences that are not significant statistically
- references are sometimes misused, sometimes missing in the reference list, the format is not always the same (capital letters sometimes, misspelling of some names, wrong date) and the reference list contains references that are not cited in the manuscript.
- the formatting of the section title is sometimes wrong (“.Results”)
See hereunder specific suggestions and comments:
Abstract
L22: "it is" or is
L33: "did not"
L36: "was not"
L36: two
Introduction
L46: missing ")" after reference
L51-60: the information in this paragraph are correct but it is rather unclear.
I advise to re-word it:
"Lairage is a critical stage in the swine production chain since it can negatively affect animal welfare (ÄŒobanović et al., 2016a) and cause irreparable damage to the carcass and meat quality, compromising the good practices efforts in previous stages (Faucitano, 2010). Furthermore, there is a reduced opportunity for animals to recover from stress caused by loading, transport and unloading (García-Celdrán et al., 2012; Jama et al., 2016). Skin lesions can act as an animal welfare (Carroll et al., 2018) and meat quality indicator (Guàrdia et al., 2009). Also, they might be associated with negative interactions between animals as well as their body location may correlate with the type of behavior established (Turner et al., 2006)."
In particular:
- "Furthermore, it must be considered that poorer meat quality can be associated with damages caused in pigs." : If you do not have references and more details (i.e. which meat quality indicators are associated with lesions?) then I would delete this sentence, as it does not convey more information than what you say in the previous and following sentences.
- Please provide more information on the "type of behaviour established": what do you mean and which behaviours?
L61: misuse of "thereby" - it is usually used in the middle of a sentence to refer to a previous statement.
I recommend changing it for "consequently", as I imagine this is what you meant.
L63: I guess you meant that "slaughter assessments are an important tool as it allows to sample a large number of animals per day"
"viability" does not apply here...
Material and Methods
L76: "...of approximately 5 month old"
L85: missing "." between sentences
L88: "on the day scheduled"
L106: "weighing area" (missing space)
L111: Please detail where the skin lesions assessment were conducted.
L113: I suggest rewording for clarity : "...which had two 170 cm long, non-branched, metal chains equally spaced and hanging in the middle of the pen 30 cm above the floor". Also, please add the distance between the chains (I could only see two chains on the picture provided, so I assumed you divided the pen in three and hanged the chains at 1/3 and 2/3 of the pen)?
L115: were not
Table 1:
- missing spaces:
Number of assessed pigs
stocking density at lairage
Dead at arrival
- what is the unit of lairage duration ?
L123: stocking density (missing space)
L133: I am not sure "calibrated" applies to humans... I guess you trained the observers beforehand and made sure they had a good agreement in their observations? If so, please provide the inter-observer reliability score.
Also, please write fully numbers lower than 10 that are not units of measures (e.g. here : "two trained observers"
L134: Do you mean that each observer monitored only one pen ? If so reword: "where each one was responsible for monitoring one of the two pens (NE or EE)"
L144: why was only the left side of pigs assessed ?
L168: This is the first time you refer to "A2" without stating its meaning, I understood later that this referred to "assessment 2 (after slaughter)". Please state the meaning of A2 here.
L170-171: Arguable: lesions in the back could be caused by bullying (biting) from other pigs
Figure 5: is the aim of this picture to show the three types of lesions too? If so, this should be indicated in the caption or under each picture, as you did for Figure 4
L187: presented in Table 2
L188: do you mean that the proportions were calculated relative to the number of pigs per pen (i.e. total number of behaviours observed/number of pigs) ?
L189: "two categories"
Table 2 : missing spaces:
"stationary or walking"
"body supported"
"surface supported"
"another pig"
I would not use "supported on the floor" (which is grammatically incorrect as something is "suppoted by" and not "supported on", and sementically incorrect as the body is not per se supported by the floor) but rather "in contact with the floor"
L213: "arcsine of the square root"
L213-214: "model requirements" not "suppositions" and please detail what other requirements were checked.
Results
L225: "there was none isolated effect" or "there was not an isolated effect"
Table 3: footnote is not relevant as p-values are written in full, please remove
L239: There is a misuse of the citations in this sentence: Staaveren et al., 2015 and Carroll et al., 2018 (if that was the intended citation, as "Carrol et al., 2015" is not in the reference list) did not observe LESS lesions in their slaughter assessment but MORE, as the visibility was INCREASED by the slaughtering process (hair removal, cleaning of skin, better light and no movement from the animals). Therefore, you should remove this statement (or at least the references, and your should explain which aspects of technological slaughter operations hindered the visibility of lesions in your case)
That is, however, good references to justify that more lesions were found between A1 and A2 (besides the effects of lairage)
Figure 6: Please display error bars and the significant differences, if any were detected.
244-264: as no significant differences were found, I would advise the authors to remove wording such as "greater"; "lower" and "unexpectedly", which are usually used to emphasise results that are supported by the statistical analyses. It is ok to reported numbers observed, but I think a table, or figures (with error bars please), would be sufficient and clearer than a paragraph that suggest to the reader that statistical differences were found.
L270: was not
Table 4: Footnote 2 is not relevant as the p-values are written in full.
L283-297: was any of this significant statistically ? if not, state that no difference were found and refer to the Figure. This is again quite misleading for the reader who might understand that significant differences were found. If significant differences were found, then state it (p-value etc).
L286: did not
Figure 7: please manage the axis titles and chart titles better, some are only partly readable.
L306, L307 and L309: did not
Table 5: again, the footnote about significance is not relevant as p-values are written in full
L319: higher behaviour proportions
L319-341: is any of that supported by the statistics?! If not, you should not present it in the result section. It is fine to discuss numerical differences in the discussion, for the sake of debate, but not in the results.
You should remove every allegation ("lower", "increased", "reduced", "higher", etc) not supported by the statistics, and include them in the discussion, if you wish.
Discussion
L356: Pérez et al 2002 is not in the list of references
L357: I do not understand, your lairage period was over 4 h, wasn't it ? So, which experience do you refer to ? And how long is the resting period in commercial conditions ?
Or is the resting period a period before lairage ?
L360: did not
L359-364: I do not understand the aim of this paragraph, why do you refer to two studies that contradicts what you found and what many others found ? Morever, one of them did not mix the pigs, which (as you say yourself) is likely to have contributed to non-increase in lesions...
L365-371: This paragraph fits better in the introduction as it introduces the novelty aspect of your study... I suggest to move it there
L375: did not
L379: misuse of thereby (see previous comment), did you mean "therefore"?
L380-382: yes something is better than nothing, but actually many studies found that chains are not great compared to other (more accessible, made of natural material) enrichment.
L385, 388, 391 and 393: did not, were not, did not, did not
L397-399: Yes, but that was not significant in your case, you need to recall that
L408: were not
L406-409: good point, I think it would help also to state how many animals were in the pen (so how many animals/chain) so that the reader can understand how accessible the chains were to the animals (in addition, the stocking density of animals/m2, instead of kg/m2 would help better to visualize how crowded the pen was)
Conclusion
L426: I do not understand the meaning of this sentence...
L433: could not
L434-436: inconsistencies in the names of authors (sometimes initials, sometimes full name)
References not used: 3, 4, 9, 15, 26, 39 and 42
Comments on the Quality of English LanguageSee in general comments
Author Response
Abstract
L22: "it is" or is
Response: Reviewed
L33: "did not"
Response: Reviewed
L36: "was not"
Response: Reviewed
L36: two
Response: Reviewed
Introduction
L46: missing ")" after reference
Response: Reviewed
L51-60: the information in this paragraph are correct but it is rather unclear.
I advise to re-word it:
"Lairage is a critical stage in the swine production chain since it can negatively affect animal welfare (ÄŒobanović et al., 2016a) and cause irreparable damage to the carcass and meat quality, compromising the good practices efforts in previous stages (Faucitano, 2010). Furthermore, there is a reduced opportunity for animals to recover from stress caused by loading, transport and unloading (García-Celdrán et al., 2012; Jama et al., 2016). Skin lesions can act as an animal welfare (Carroll et al., 2018) and meat quality indicator (Guàrdia et al., 2009). Also, they might be associated with negative interactions between animals as well as their body location may correlate with the type of behavior established (Turner et al., 2006)."
In particular:
- "Furthermore, it must be considered that poorer meat quality can be associated with damages caused in pigs." : If you do not have references and more details (i.e. which meat quality indicators are associated with lesions?) then I would delete this sentence, as it does not convey more information than what you say in the previous and following sentences.
Response: Reviewed
Lairage time in the slaughterhouse is a critical stage in the swine production chain since it can negatively affect animal welfare (ÄŒobanović et al., 2016a) and cause irreparable damage to the carcass and meat quality, compromising the good practices efforts in previous stages (Faucitano, 2010) and reducing the opportunity for animals to recover from stress caused by loading, transport and unloading (García-Celdrán et al., 2012; Jama et al., 2016). Furthermore, it must be considered that poorer meat quality can be associated with fights between pigs, mainly due to transport stress and other factors such as batch mixtures and dominance (Sobral et al., 2024). Thus, skin lesions can act as an animal welfare (Carroll et al., 2018) and meat quality indicator (Guàrdia et al., 2009).
- Please provide more information on the "type of behaviour established": what do you mean and which behaviours?
Response: Reviewed
Sentence deleted!
L61: misuse of "thereby" - it is usually used in the middle of a sentence to refer to a previous statement.
I recommend changing it for "consequently", as I imagine this is what you meant.
Response: Reviewed
Consequently, several approaches to monitor animal welfare conditions have been investigated by the scientific community.
L63: I guess you meant that "slaughter assessments are an important tool as it allows to sample a large number of animals per day"
"viability" does not apply here...
Response: Reviewed
Slaughter assessments are an important as it allows to sample a large of animals per day
Material and Methods
L76: "...of approximately 5 month old"
Response: Reviewed
L85: missing "." between sentences
Response: Reviewed
L88: "on the day scheduled"
Response: Reviewed
L106: "weighing area" (missing space)
Response: Reviewed
L111: Please detail where the skin lesions assessment were conducted.
L113: I suggest rewording for clarity : "...which had two 170 cm long, non-branched, metal chains equally spaced and hanging in the middle of the pen 30 cm above the floor". Also, please add the distance between the chains (I could only see two chains on the picture provided, so I assumed you divided the pen in three and hanged the chains at 1/3 and 2/3 of the pen)?
Response: Reviewed
After skin lesion assessment, half of each group of animals was randomly conducted to the conventional lairage pen (NE) and the other half to the enriched lairage pen (EE), which had non-branched metal chains, equally spaced metal chains hanging in the middle. of the pen at a height of 2.0 m with a distance of 30 cm from the floor, distributed in a central row with 3 units (Figure 2).
L115: were not
Table 1:
- missing spaces:
Number of assessed pigs
stocking density at lairage
Dead at arrival
- what is the unit of lairage duration ?
Response: Reviewed
L123: stocking density (missing space)
Response: Reviewed
L133: I am not sure "calibrated" applies to humans... I guess you trained the observers beforehand and made sure they had a good agreement in their observations? If so, please provide the inter-observer reliability score.
Also, please write fully numbers lower than 10 that are not units of measures (e.g. here : "two trained observers"
Response: Reviewed
L134: Do you mean that each observer monitored only one pen ? If so reword: "where each one was responsible for monitoring one of the two pens (NE or EE)"
Response: Reviewed
At lairage, animal behaviour was monitored once the pens NE and EE were fully loaded during the first 4 hours of the resting period by 2 trained and calibrated observers, where each one was responsible for monitoring one of the two pens (NE or EE).
L144: why was only the left side of pigs assessed ?
Response: Protocol do Welfare Quality of Pigs
L168: This is the first time you refer to "A2" without stating its meaning, I understood later that this referred to "assessment 2 (after slaughter)". Please state the meaning of A2 here.
Response: Reviewed
L170-171: Arguable: lesions in the back could be caused by bullying (biting) from other pigs
Figure 5: is the aim of this picture to show the three types of lesions too? If so, this should be indicated in the caption or under each picture, as you did for Figure 4
Response: The figures had their titles changed! Figure 5 was used as a standard for classifying animals according to injuries. Therefore, the other figure was excluded.
L187: presented in Table 2
Response: Reviewed
L188: do you mean that the proportions were calculated relative to the number of pigs per pen (i.e. total number of behaviours observed/number of pigs) ?
Response: Yes
L189: "two categories"
Response: Reviewed
Table 2 : missing spaces:
"stationary or walking"
"body supported"
"surface supported"
"another pig"
I would not use "supported on the floor" (which is grammatically incorrect as something is "suppoted by" and not "supported on", and sementically incorrect as the body is not per se supported by the floor) but rather "in contact with the floor"
Table 2 - Response: Reviewed
L213: "arcsine of the square root"
Response: Reviewed
L213-214: "model requirements" not "suppositions" and please detail what other requirements were checked.
Response: Reviewed
Results
L225: "there was none isolated effect" or "there was not an isolated effect"
Response: Reviewed
Table 3: footnote is not relevant as p-values are written in full, please remove
Response: Reviewed
L239: There is a misuse of the citations in this sentence: Staaveren et al., 2015 and Carroll et al., 2018 (if that was the intended citation, as "Carrol et al., 2015" is not in the reference list) did not observe LESS lesions in their slaughter assessment but MORE, as the visibility was INCREASED by the slaughtering process (hair removal, cleaning of skin, better light and no movement from the animals). Therefore, you should remove this statement (or at least the references, and your should explain which aspects of technological slaughter operations hindered the visibility of lesions in your case)
That is, however, good references to justify that more lesions were found between A1 and A2 (besides the effects of lairage)
Response: This justification was removed, as the session is about results!
Figure 6: Please display error bars and the significant differences, if any were detected.
244-264: as no significant differences were found, I would advise the authors to remove wording such as "greater"; "lower" and "unexpectedly", which are usually used to emphasise results that are supported by the statistical analyses. It is ok to reported numbers observed, but I think a table, or figures (with error bars please), would be sufficient and clearer than a paragraph that suggest to the reader that statistical differences were found.
L270: was not
Response: Reviewed
Table 4: Footnote 2 is not relevant as the p-values are written in full.
Response: Reviewed
L283-297: was any of this significant statistically? if not, state that no difference were found and refer to the Figure. This is again quite misleading for the reader who might understand that significant differences were found. If significant differences were found, then state it (p-value etc).
L286: did not
Response: Reviewed
Figure 7: please manage the axis titles and chart titles better, some are only partly readable.
L306, L307 and L309: did not
Response: Reviewed
Table 5: again, the footnote about significance is not relevant as p-values are written in full
Response: Reviewed
L319: higher behaviour proportions
Response: Reviewed
L319-341: is any of that supported by the statistics?! If not, you should not present it in the result section. It is fine to discuss numerical differences in the discussion, for the sake of debate, but not in the results.
You should remove every allegation ("lower", "increased", "reduced", "higher", etc) not supported by the statistics, and include them in the discussion, if you wish.
Response: Reviewed
Discussion
L356: Pérez et al 2002 is not in the list of references
Pérez, M. P.; Palacio, J.; santolaria, M. P.; et al. Influence of lairage time on some welfare and meat quality parameters in pigs. Veterinary Research, v. 33, n. 3, p. 239-250, 2002.
L357: I do not understand, your lairage period was over 4 h, wasn't it? So, which experience do you refer to? And how long is the resting period in commercial conditions?
Or is the resting period a period before lairage ?
Reponse: The resting period is the same as the lairage period, these are terms that have meaning! The lairage period was up to 4 hours!
L360: did not
Response: Reviewed
Deleted: “unlike this experiment based on commercial conditions”
L359-364: I do not understand the aim of this paragraph, why do you refer to two studies that contradicts what you found and what many others found? Morever, one of them did not mix the pigs, which (as you say yourself) is likely to have contributed to non-increase in lesions...
Response: Thanks for the comment. We included these studies because they are the few that were carried out with environmental enrichment in slaughterhouses. The results are interesting and we can make analogies with our results. Furthermore, with different objectives used in our study! As discussed below!
L365-371: This paragraph fits better in the introduction as it introduces the novelty aspect of your study... I suggest to move it there
Response: Reviewed
L375: did not
Response: Reviewed
L379: misuse of thereby (see previous comment), did you mean "therefore"?
Response: Reviewed
L380-382: yes something is better than nothing, but actually many studies found that chains are not great compared to other (more accessible, made of natural material) enrichment.
L385, 388, 391 and 393: did not, were not, did not, did not
L397-399: Yes, but that was not significant in your case, you need to recall that
Response: In general, those findings corroborate the data presented in this experiment, which negative interaction frequencies were lower in groups submitted to EE lairage, although not statistically significant with the group enriched with NE.
L408: were not
Response: Reviewed
L406-409: good point, I think it would help also to state how many animals were in the pen (so how many animals/chain) so that the reader can understand how accessible the chains were to the animals (in addition, the stocking density of animals/m2, instead of kg/m2 would help better to visualize how crowded the pen was)
Conclusion
L426: I do not understand the meaning of this sentence...
Response: Reviewed
L433: could not
Response: Reviewed
L434-436: inconsistencies in the names of authors (sometimes initials, sometimes full name)
Response: Reviewed
References not used: 3, 4, 9, 15, 26, 39 and 42
Response: Deleted
Round 2
Reviewer 4 Report
Comments and Suggestions for Authors
the paper improved a lot
Author Response
ok!
Reviewer 5 Report
Comments and Suggestions for Authors
I thank the authors for this nicely revised manuscript!
I just spotted one typo at L256 : there was no influence (instead of "there was not influence")
Author Response
Revised